# Bioleaching of Phosphate Minerals Using *Aspergillus niger*: Recovery of Copper and Rare Earth Elements

**Laura Castro \*, Maria Luisa Blázquez, Felisa González and Jesús Angel Muñoz**

Department of Chemical and Materials Engineering, Complutense University of Madrid, 28040 Madrid, Spain; mlblazquez@quim.ucm.es (M.L.B.); fgonzalezg@quim.ucm.es (F.G.); jamunoz@quim.ucm.es (J.A.M.)

\* Correspondence: lcastror@ucm.es; Tel.: +34-91-394-4354

**Abstract:** Rare earth elements (REE) are essential in high-technology and environmental applications, where their importance and demand have grown enormously over the past decades. Many lanthanide and actinide minerals in nature are phosphates. Minerals like monazite occur in small concentrations in common rocks that resist weathering. Turquoise is a hydrous phosphate of copper and aluminum scarcely studied as copper ore. Phosphate-solubilizing microorganisms are able to transform insoluble phosphate into a more soluble form which directly and/or indirectly contributes to their metabolism. In this study, bioleaching of heavy metals from phosphate minerals by using the fungus *Aspergillus niger* was investigated. Bioleaching experiments were examined in batch cultures with different mineral phosphates: aluminum phosphate (commercial), turquoise, and monazite (natural minerals). The experiments were performed at 1% pulp density and the phosphorous leaching yield was aluminum phosphate > turquoise > monazite. Bioleaching experiments with turquoise showed that *A. niger* was able to reach 8.81 mg/l of copper in the aqueous phase. Furthermore, the fungus dissolved the aluminum cerium phosphate hydroxide in monazite, reaching up to 1.37 mg/L of REE when the fungus was grown with the mineral as the sole phosphorous source. Furthermore, *A. niger* is involved in the formation of secondary minerals, such as copper and REE oxalates.

**Keywords:** *Aspergillus niger*; mineral phosphate; bioleaching; rare earth elements

## 1. Introduction

Rare Earth Elements (REE) attract enormous interest because of their importance to current and high technological manufacturing industries and the expectation of an increase in requirements throughout the world on a monopolistic market of REE due to geopolitical controls [1]. The distribution of rare earths by their end use is estimated as follows: catalysts, 60%; ceramics and glass, 15%; metallurgical applications and alloys, 10%; polishing, 10%; others, 5% [2].

The production of rare earths increased with renewed mining activities in the United States of America as well as new and or increased production in Australia, Burma (Myanmar), and Burundi. Mine production in China increased 14% in 2018 compared with the quota in 2017. In 2018, the USA Department of the Interior and other executive branch agencies published a list of 35 critical minerals, including rare earths [3]. Likewise, the European Commission carries out a critical assessment on non-energy and non-agricultural raw materials that, in 2017, included heavy rare earth elements, light rare earth elements, and platinum group metals [4].

Despite their name, these elements are not rare and occur abundantly in the Earth's crust, however, not at high enough concentrations to be considered economically recoverable. The elements vary in abundance from cerium, the 25th most abundant element of common elements at 60 parts per million,

to thulium and lutetium, the least abundant rare earth elements in the Earth at about 0.5 ppm [5]. The three main REE ores that are currently mined for production are bastnaesite (REE-FCO$_3$), monazite (light REE-PO$_4$), and xenotime (heavy REE-PO$_4$), together representing approximately 95% of known REE minerals. Conventional methods of REE extraction from ores involve the use of high temperatures and harsh chemicals (either concentrated sulfuric acid or concentrated NaOH), resulting in high energy usage and the production of toxic waste streams [6].

Biohydrometallurgical methods are generally considered as an eco-friendly alternative with low cost and low energy requirements. Bioleaching allows the recovery of some valuable heavy metals and reduces the toxicity of the waste materials using microorganisms [7,8]. Recently, some research work has been developed, especially with monazite, using microorganisms able to dissolve phosphorus from inorganic rocks, named as phosphate-solubilizing microorganisms (PSMs). PSMs have been previously used in agriculture to enhance crop production, but there are few studies related to the recovery of valuable elements from phosphate minerals. Several microorganisms, such as the bacteria *Enterobacter aerogenes*, *Pantoea agglomerans*, and *Pseudomonas putida*, have demonstrated to grow in the presence of natural rare earth phosphate minerals, releasing phosphorous, iron, thorium, and REE. Numerous organic acids were produced by the microorganisms, but microbial processes were also playing a role in solubilization of the monazite ore [9]. *Azospirillum brasilense*, *Azospirillum lipoferum*, *Pseudomonas rhizosphaerae*, and *Mesorhizobium cicero*, but particularly *Acetobacter aceti*, were able to solubilize cerium and lanthanum from monazite, although the efficiency of the process was low [10].

Fungal strains able to solubilize phosphate minerals have been also used to leach monazite releasing rare earth elements to the aqueous phase, such as *Aspergillus niger* ATCC 1015, *Aspergillus terreus* strain ML3-1, and a *Paecilomyces* spp. strain WE3-F [11]. The fungus *Penicillum* sp. solubilized a total concentration of 12.32 mg L$^{-1}$ rare earth elements after 8 days. Although monazite also contains radioactive thorium, bioleaching by these fungi preferentially solubilized rare earth elements over thorium that remained in the solid residual [9]. The most important mechanism of metal leaching by heterotrophic fungus is the production of organic acids, however, phosphatase activity and the microbial attachment could be involved in the dissolution of monazite [12].

*Aspergillus* species have shown potential for rare earth metals bioleaching of various waste materials, such as fly ash [13], spent catalysts [14], and electrical waste [15]. Some of these processes reached leaching recoveries similar to the yields offered by chemical leaching.

Copper has been extensively used in industry due to its excellent ductility and electric and thermal conductivity. Copper bioleaching has commercial application in the metal extraction from low grade and secondary mineral resources, mainly sulfides [16,17]. Nevertheless, copper phosphate bioleaching has been scarcely investigated [18].

Several hundred phosphate minerals are known and the most important mineral is apatite. Apatites and poorly crystalline or amorphous phosphorite sediments are mined and treated with acids to obtain fertilizers. Nevertheless, phosphates have been scarcely mined to extract valuable metals. Many of the f-block elements are extracted from phosphate minerals, notably monazite [19]. Turquoise mineral was considered in this study as a source of copper. Some copper companies do not consider turquoise valuable enough for sale; instead, the mineral is crushed and processed for its metal content. The main objective of the present work is to study the solubilization of phosphorous and the recovery of valuable elements from mineral phosphates (aluminum phosphate, turquoise, and monazite) using the fungus *A. niger*. The release of copper from turquoise and rare earth elements (Ce, La, and Nd) from monazite was monitored. In addition, the pH and the redox potential (Eh$_{Ag/AgCl}$) were followed during the process, evidencing that the mineral dissolution is caused by the fungal growth and the production of organic acids that acidified the medium and complexed the metals. The samples were examined by X-ray diffraction (XRD) and Scanning Electron Microscopy (SEM), showing that *A. niger* metabolites are involved in the formation of secondary minerals, such as copper and REE-oxalates.

## 2. Materials and Methods

### 2.1. Minerals

Different phosphate minerals, both commercial and natural, were used in the bioleaching tests (Table 1).

**Table 1.** Mineral phosphates tested in the bioleaching experiments using *A. niger*.

| Mineral | Formula | Type | Origin |
|---------|---------|------|--------|
| Berlinite | $AlPO_4$ | Commercial | Fluka |
| Turquoise | $CuAl_6(PO_4)_4(OH)_3 \cdot 5H_2O$ | Natural | Mirandilla, Badajoz (Spain) |
| Monazite | $(Ce, La, Nd, Th)PO_4$ | Natural | Seis Lagos (Brazil) |

The elemental composition of the natural phosphate minerals was determined by X-ray fluorescence chemical analysis. The composition of the turquoise was (wt. %): Si, 22.4; Al, 12.9; P, 7.3; Cu, 3.5; Fe, 2.0; O, 48.9. The main elements in the monazite were (wt. %): Al, 16.7; Si, 16.5; Fe, 4.95; S, 4.59; P, 1.42; Ce, 1.23; Nd, 0.69; La, 0.62; Th, 0.54; O, 46.8.

### 2.2. Leaching Fungal Strain and Bioleaching Experiments

The fungus *Aspergillus niger* CECT2807 was provided by the Spanish Type Culture Collection.

The bioleaching experiments were carried out in 250 mL Erlenmeyer flasks in triplicate. Each flask containing 1% mineral (*w/v*) was autoclaved for 30 min at 121 °C. No mineralogical changes were observed in X-ray diffractograms before and after autoclaving. A volume of 100 mL of sterile potato dextrose broth (potato starch (from infusion) 4 g/L, dextrose 20 g/L, Difco, Carlsbad, CA, USA) at initial pH 5.3 was added to each flask. Flasks were inoculated with the fungus *A. niger*, and then, cultures were incubated aerobically on an orbital shaker (New Brunswick Scientific Innova 44R) at 150 rpm and 30 °C. Non-inoculated flasks were used as controls. An amount of 5 mL was periodically withdrawn during the experiments for further analysis (phosphorous and metal concentrations, pH, and $E_{Ag/AgCl}$.

In addition, *A. niger* was grown on non-sterile monazite and sterile potato dextrose broth to observe the influence of indigenous biota on the bioleaching process. The fungus was also grown with monazite as the sole phosphate source on modified minimal medium containing (g/L): $NaNO_3$, 6.0; KC1, 0.52; $MgSO_4 \cdot 7H_2O$, 0.52; glucose, 10. All the reagents were provided by Fisher Chemicals (percent purity ≥99%).

### 2.3. Analytical Methods

pH and redox potential ($Eh_{Ag/AgCl}$) of the samples were measured using a pHmeter Crison Basic 20 (sensitivity: 98%) (Hach Lange., Barcelona, Spain).

The concentration of phosphorous was determined using an ICP-OES equipment (Perkin Elmer Optima 2100 DV, Perkin Elmer Inc., Waltham, MA, USA). In addition, the concentration of copper during turquoise experiments and cerium, lanthanum, and neodymium concentrations in monazite experiments were measured by ICP-OES. Each measurement was performed in triplicate with RSD <3%.

The morphology of fungi and the different phosphate minerals was observed using Scanning Electron Microscopy (SEM, JEOL JSM-6330 F, JEOL, Boerne, TX, USA). Samples with fungus were prepared by filtering through 0.2 μm pore-size filters and successively dehydrated with acetone/water mixtures of 30%, 50%, and 70% acetone, respectively, and stored overnight at 4 °C in 90% acetone for cell dehydration. After critical-point drying and coating with gold, samples were observed under the microscope FE-SEM at 15–20 kV.

Mineral residues in culture flasks were characterized using powder X-ray diffraction (XRD) on a Philips X'pert-MPD equipment (Malvern Panalytical, Eindhoven, The Netherlands) with a Cu anode operating at a wavelength of 1.5406 Å as the radiance source. Samples were placed on off-axis quartz

plates (18 mm diameter × 0.5 mm DP cavity). The scanning range was from 10° to 90° 2θ, with an angular interval of 0.05° and 4 s counting time. The crystalline phases were identified using standard cards from the International Centre for Diffraction Data (ICDD, Newtown Square, Pennsylvania) Powder Diffraction File database.

## 3. Results

### 3.1. Phosphate Mineral Solubilization

Phosphorous is an essential macronutrient for all living organisms. This element has a fundamental role in biochemical reactions involving genetic material (DNA, RNA) and energy transfer (ATP), and in the formation of cell membranes (phospholipids) that provide structural support for organisms [20]. Consequently, phosphorous is a limiting nutrient for microorganisms in many environments due to the deficit of organophosphates or the low solubility of inorganic mineral phosphates (usually present in soils and rocks). Microorganisms have to produce enzymes named phosphatases or generate metabolic products such as organic acids to dissolve the insoluble phosphorous immobilized in the crystalline structure of some minerals [21].

Calcium phosphates can be solubilized by a wide range of microorganisms. Nevertheless, there are other valuable phosphate minerals which are more refractory. The fungus *Aspergillus niger* CECT2807 was grown in the presence of aluminum phosphate, turquoise, and monazite and phosphorous concentration was measured periodically. The initial concentration of phosphorous in the medium was 27.63 mg/L. Aluminum phosphate was abiotically dissolved, but *A. niger* enhanced the phosphorous solubilization rate. When the fungus strain was grown with turquoise, phosphorous concentration decreased to 5.10 mg/L after 7 days, and then, the concentration increased, reaching 47.97 mg/L after 14 days. *A. niger* also grew in presence of monazite, as shown in Figure 1. After 7 days, phosphorous concentration decreased up to 0.70 mg/L, and then, increased up to 8.81 mg/L.

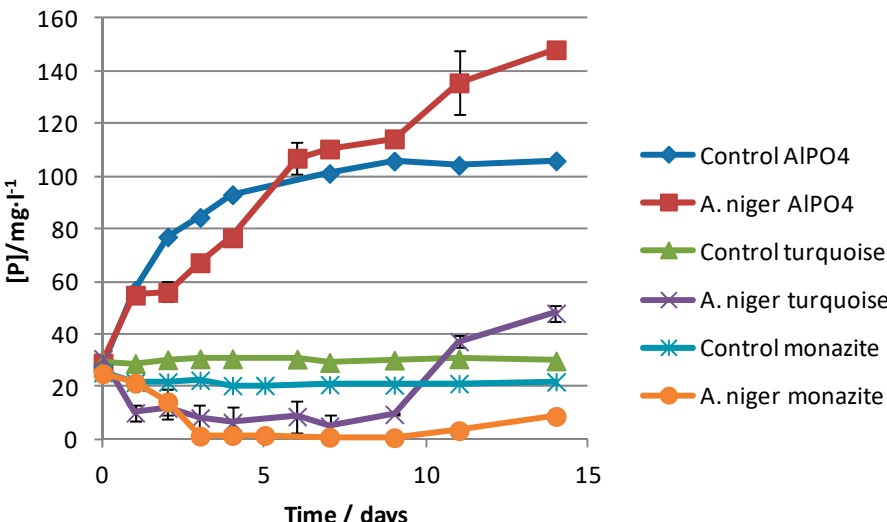

**Figure 1.** Evolution of phosphorous concentration from different mineral phosphates (aluminum phosphate, turquoise and monazite) at 1 g/L in abiotic and biotic (*A. niger*) tests.

*A. niger* was able to transform remarkably two very insoluble minerals, turquoise and monazite, with an acidification of the medium (Figure 2a). The proton substitution reaction is responsible for the mineral phosphate solubilization through the biological production of organic acids, which could be represented by the following general Equation [22]:

$$[M^{n+}][PO_4{}^{3-}] + [HA] = [H^+] \, [PO_4{}^{3-}] + [M^{n+}] \, [A^-] \tag{1}$$

where [HA] is the corresponding organic acid and $M^{n+}$ is a metallic cation. The stoichiometry is not included in this equation because it depends on the mineral phosphate and the type of organic acid generated by the microorganism. The efficiency of solubilization depends on the mineral phosphate solubility, the amount of protons generated by the organic acid, and its pKa.

After 7 days, the pH values in the inoculated flasks with $AlPO_4$ and turquoise experienced a sharp increase associated with the release of phosphorous in the aqueous phase due to proton consumption. In addition, turquoise is a hydrated basic phosphate that raises the $OH^-$ concentration during its solubilization.

The redox potential was also monitored during the bioleaching processes (Figure 2b). The increase in the redox potential observed at the initial stage, in parallel to the decrease in the pH value in the flasks, was attributed to the solubilization of mineral phosphates by the fungus via direct oxidation of glucose to gluconic acid and other organic acids.

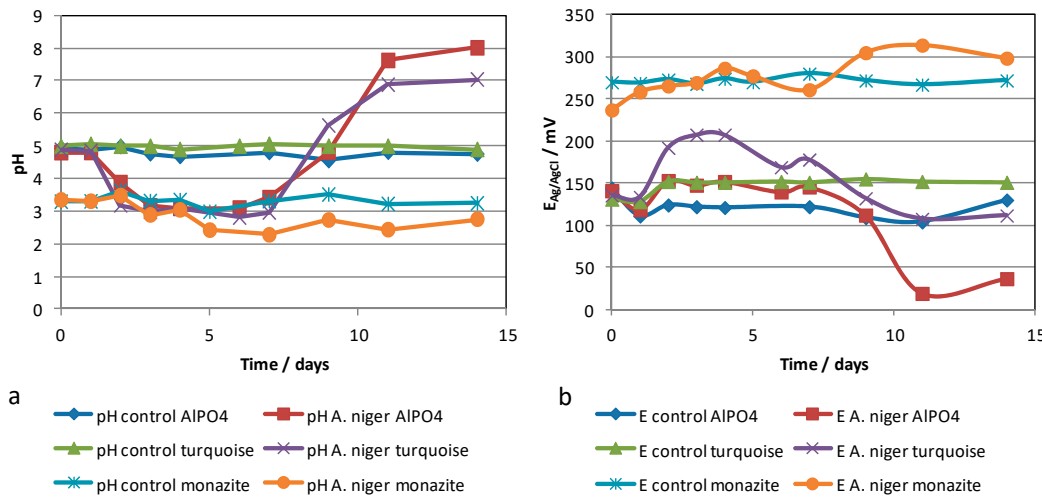

**Figure 2.** Evolution of (**a**) pH and (**b**) $E_{Ag/AgCl}$ in abiotic and biotic (*A. niger*) tests for different mineral phosphates (aluminum phosphate, turquoise, and monazite) at 1 g/L.

Microbial bioleaching of insoluble metal phosphate minerals is caused by protonation of the anion of the metal compound. In fungi, two mechanisms seem to be the most relevant for solubilization of phosphate minerals: (1) the $H^+$-translocating ATPase of the plasma membrane that is a source of protons, and (2) the production of organic acids, which can be effective complexing agents for metal cations [21]. In addition, the excretion of metabolites, siderophores, and $CO_2$ from microbial respiration also contributes to the dissolution process [23]. The importance of all these processes varies depending on the microorganism and the growth conditions. Previous research has evidenced that *A. niger* produces citric acid when it is grown with metal phosphates [24]. Organic acids in the soil promote the chelation of cations and favor anions release, such as $PO_4{}^{3-}$, leading to the solubilization of phosphate compounds.

### 3.2. Turquoise Bioleaching using Aspergillus niger

There are no previous studies on turquoise bioleaching to recover copper. The evolution of copper concentration in the aqueous phase during the dissolution of turquoise by fungal action is shown in Figure 3. Copper concentration in the aqueous phase was 8.81 mg/L after 14 days. However, the maximum concentration was 13.02 mg/L, reached after 9 days. At that moment, the pH increased and the redox potential decreased as a consequence of copper precipitation (Figure 2).

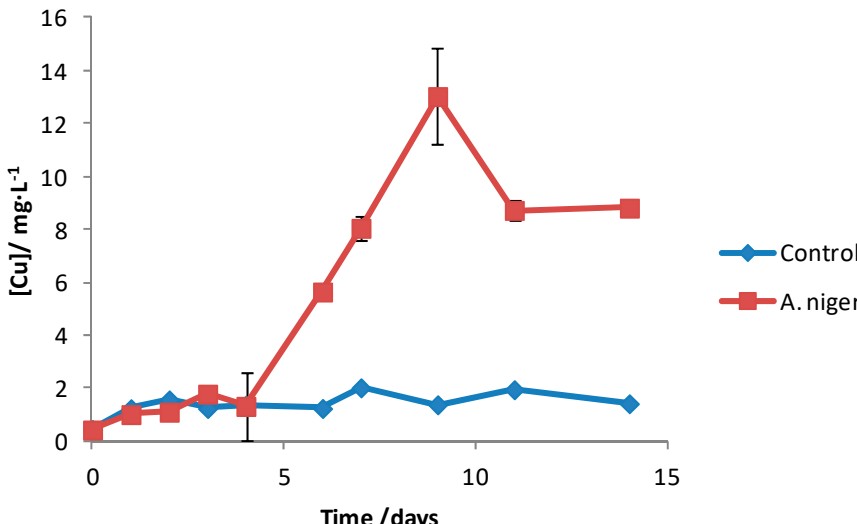

**Figure 3.** Evolution of copper concentration from turquoise (1 g/L) in abiotic and biotic (*A. niger*) tests.

Organic acids, citric, oxalic and gluconic acid, are generated by the *A. niger*, whether a mineral phosphate is present or not [25]. The solubilization of insoluble metal phosphates release free metal cations and/or soluble metal complexes, increasing their availability. Attending to their high solubility and the binding constants, gluconate ($[CuL^+]$, $7.1 \times 10^7$; $[CuL_2]$, $2.5 \times 10^{14}$) and citrate ($[CuL]^-$, $1 \times 10^{18}$; $[CuHL]$, $2 \times 10^{22}$; $[CuH_2L]^+$, $2 \times 10^{28}$) would be involved in the high copper concentration in solution.

The decrease in copper concentration could be caused by the precipitation of organic metabolites [26] or by mineralogical changes (dissolution and subsequent reprecipitation) in the mineral substrate during biological solubilization [27].

Raw turquoise and the mineral residues obtained after fungal growth were analyzed by XRD diffraction to compare mineral phases (Figure 4). The turquoise diffractogram confirmed the composition obtained by X-ray fluorescence and its high crystallinity. In addition, these XRD patterns evidenced significant changes in mineralogy in the inoculated samples. Turquoise ($CuAl_6(PO_4)_4(OH)_8 \cdot 5H_2O$) was completely transformed by *A. niger* CECT2807 14 days after inoculation. The diffractogram of the mineral residue obtained showed the presence of copper phosphate and copper oxalate, evidencing the copper reprecipitation by the biological action.

The residues from the inoculated flasks containing turquoise were observed by FE-SEM (Figure 5). Turquoise particles were transformed and very dense mycelium was formed by *A. niger*, where copper phosphate and copper oxalate crystals were precipitated and adsorbed.

Fungi play an important role in the formation of organic and inorganic secondary minerals through precipitation and deposition of crystalline compounds on and within cells walls, such as oxalates and phosphates.

Previous studies evidenced that fungi can generate metal oxalates with different chemical elements and minerals [28]. The fungus *Beauveria caledonica* can solubilize minerals containing heavy metals such as cadmium, copper, lead or zinc and form oxalates in the local microenvironment and in association with the mycelium. This fungus accumulated high amounts of copper as copper oxalate in the presence of copper phosphate [29]. *A. niger* was able to solubilize metal compounds ($ZnO$, $Zn_3(PO_4)_2$ and $Co_3(PO_4)_2$) and precipitate metal oxalates [26]. The precipitation of metal oxalates may provide a detoxification mechanism, whereby fungi can survive at high concentrations of hazardous metals.

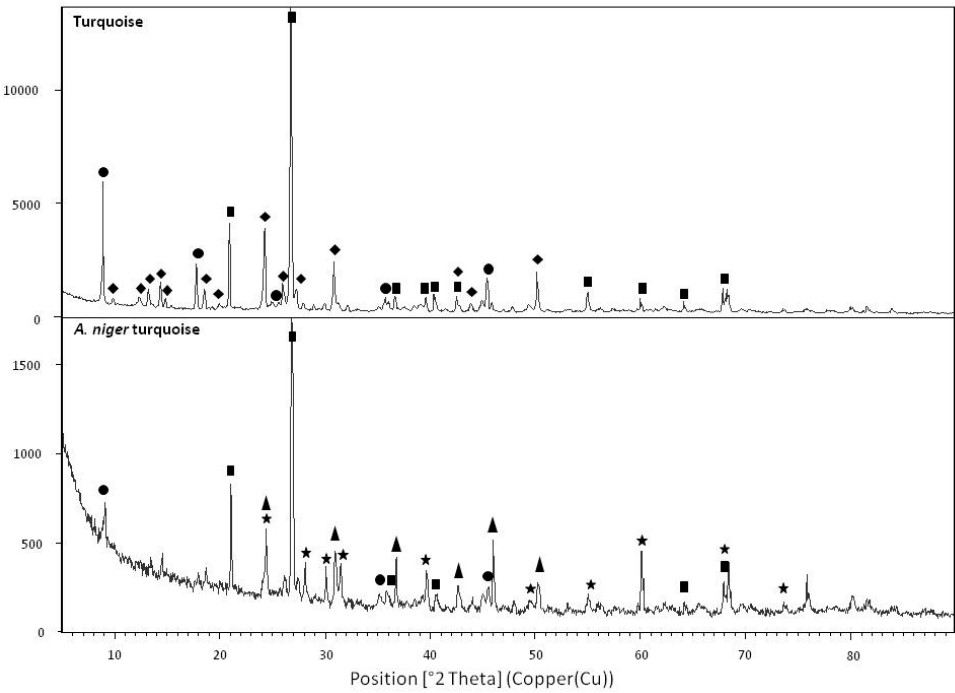

**Figure 4.** X-ray diffraction (XRD) pattern of the turquoise (**a**) before and (**b**) after 14 days of bioleaching using *A. niger* (● potassium aluminum silicate hydroxide, ■ silicon oxide, ♦ copper aluminum phosphate hydroxide hydrate, ★ copper phosphate, ▲ copper oxalate).

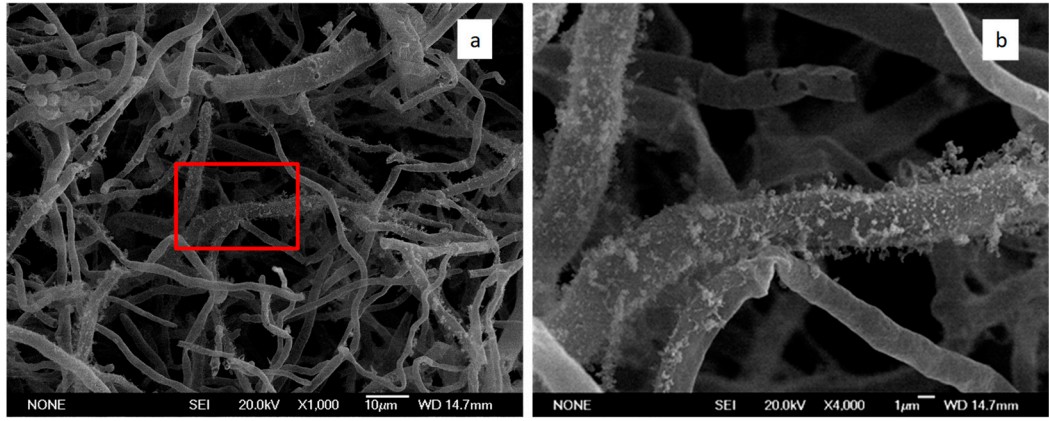

**Figure 5.** Scanning Electron Microscopy (SEM) images of (**a**) *A.niger* grown with turquoise after 14 days; (**b**) detail of precipitates on fungal biomass.

### 3.3. Monazite Bioleaching

*Aspergillus niger* CECT2807 was grown on monazite from the Morro dos Seis Lagos (Brazil). The Morro dos Seis Lagos deposit is formed by siderite as the main mineral and is covered by a thick ferruginous lateritic crust. This site hosts one of the largest Nb deposits in the world. It also contains iron and manganese and important resources of REE [30]. The evolution of REE and thorium concentrations in the aqueous phase during the bioleaching experiments with *A. niger* are shown in Figure 6. It is known that REE-phosphates have particularly low solubilities in water, in the order of $10^{-13}$ M ($10^{-11}$ g/L) [31]. After three days of bioleaching, the REE concentration reach a maximum (0.97 mg/L) corresponding to the production of higher concentrations of citric and gluconic acids by

*A. niger* (Figure 6a) [11]. A significant REE loss was observed after four days that may be caused by the removal of REE from the aqueous phase due to processes such as re-precipitation (e.g., as REE-oxalates) or adhesion to microbial cells [32,33]. Then, REE concentration began to increase, probably because the fungus needed the phosphorous from the rock for its microbial growth and phosphorous concentration changed from 0.70 mg/L at day 9 to 8.80 mg/L at day 14 (Figure 1). On the other hand, fungus biomass was able to adsorb thorium generated during the bioprocess. Other works evidenced that thorium is adsorbed onto the external cell wall by coordination with the nitrogen of the chitin present in the fungal cell wall [34,35].

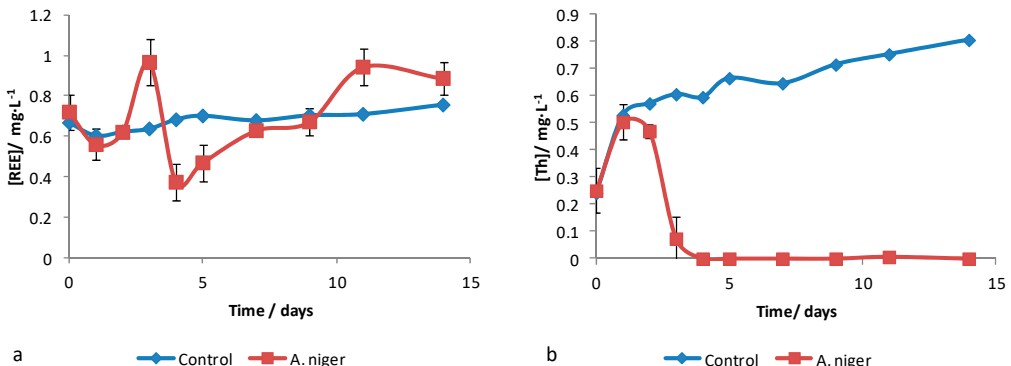

**Figure 6.** Evolution of REE (**a**) and thorium concentration released (**b**) from monazite (1 g/L) in abiotic and biotic (*A. niger*) tests.

Monazite is the main source of thorium, which is a radioactive element. Consequently, the monazite processing consumes more energy and chemicals than other REE minerals because of the requirement of additional separation of radioactive thorium. This processing generates significant amounts of hazardous wastes, leading to radioecological issues [36]. Consequently, the bioleaching of REE while thorium is adsorbed on the fungus would represent an important advantage for the development of the bioprocess.

Monazite was characterized using XRD and the main phases identified were aluminum silicate hydroxide, silicon oxide, aluminum cerium phosphate hydroxide, sodium cerium carbonate fluoride, and potassium hydronium iron sulphate hydroxide (Figure 7a). The REE ore minerals in the Morro dos Seis Lagos deposit include monazite and its alteration products, such as florencite, a REE aluminum phosphate mineral, and rhabdophane $((Ce,La)PO_4\cdot(H_2O))$ [30]. XRD also confirmed the presence of REE fluoride carbonates showing peaks corresponding to $CeCO_3F$. The mineral residue after fungal bioleaching was analyzed and the results evidenced that the aluminum cerium phosphate hydroxide was transformed by *A. niger*. Besides the mineral phases in the substrate, peaks corresponding to Ce and La oxalates were observed in samples after two weeks of incubation (Figure 7b).

Monazite samples treated with the fungus *A. niger* CECT2807 were examined by FE-SEM (Figure 8). *A. niger* developed a dense mycelium around monazite particles (Figure 8a,b). In addition, REE oxalates were precipitated by the metabolites generated during the fungal growth and adsorbed onto the mycelium (Figure 8b). The backscattered electron image showed bright areas on the filamentous structure of the fungus corresponding to REE, giving evidence of the REE oxalate precipitation and REE adsorption (Figure 8c).

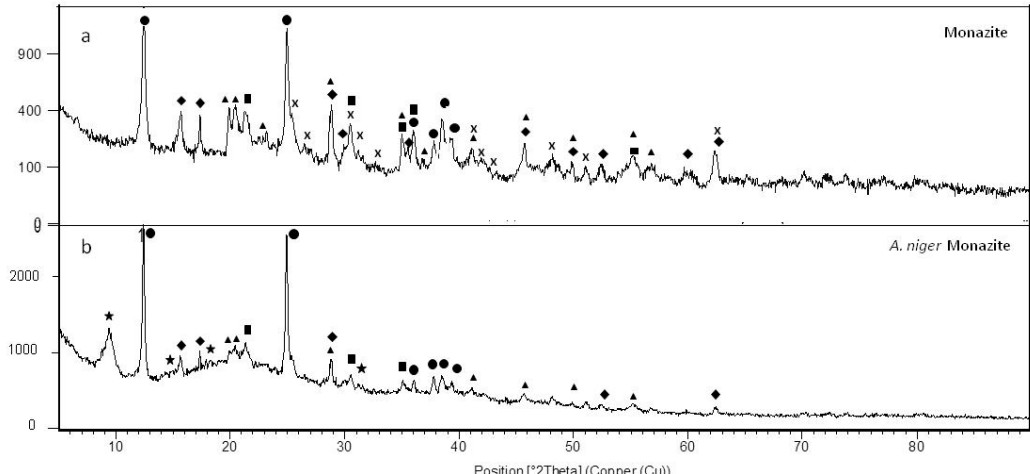

**Figure 7.** XRD pattern of the monazite (**a**) before and (**b**) after 14 days of bioleaching using *A. niger* (● aluminum silicate hydroxide, ■ silicon oxide, **x** aluminum cerium phosphate hydroxide, ▲ sodium cerium carbonate fluoride, ♦ potassium hydronium iron sulphate hydroxide, ★ cerium oxalate).

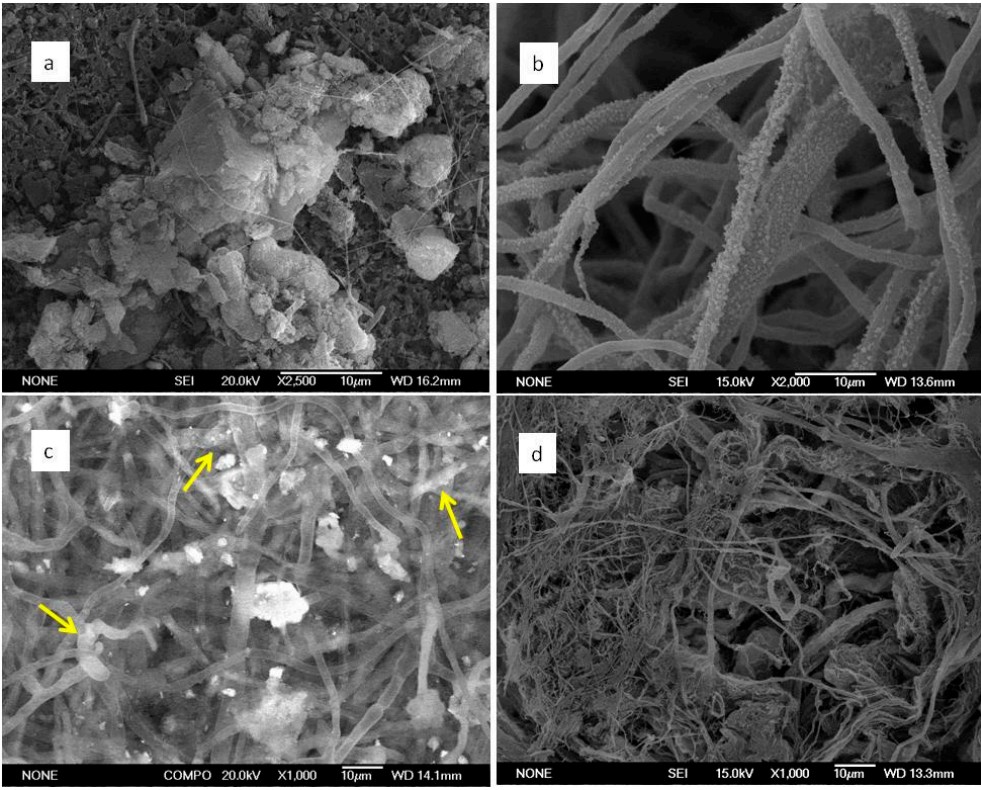

**Figure 8.** SEM images of (**a**) monazite particles after 24 h under fungal bioleaching, (**b**) mycelium developed after 14 days of fungal growth. (**c**) Backscattered electron SEM micrograph of *A. niger* mycelium. Arrows indicated bright areas on the mycelium due to REE deposition. (**d**) SEM image of mycelium encapsulating monazite mineral after 14 days of fungal growth in minimal medium.

The biotransformation of REE-bearing phosphates and the precipitation of lanthanide oxalates could follow the following equation [32]:

$$2LnPO_4 + 9H_2O + 2H^+ + 3(C_2O_4)^{2-} \rightarrow Ln_2(C_2O_4)_3 \cdot 9H_2O + 2HPO_4{}^2 \tag{2}$$

The formation of REE oxalates limits the long-term dissolution of these elements. Nevertheless, recent studies have demonstrated that biogenic REE-oxalates can be transformed through thermal decomposition into REE-oxides that may be precursors for other valuable REE materials [37].

In order to optimize the solubilization of REE from monazite by *A. niger*, different growth conditions were studied. The bioleaching of REE by *A. niger* grown on sterilized monazite with dextrose potato broth was compared with the yield obtained when the fungus was grown with a minimal medium (without phosphorous). In addition, the REE release by the indigenous microorganisms was monitored in experiments performed on non-inoculated flasks with non-sterilized monazite. Furthermore, *A. niger* was inoculated on non-sterilized monazite with the aim of evaluating a possible syntrophic effect between the microbial populations (Figure 9).

When *A. niger* was grown on dextrose potato broth with sterilized monazite, the REE concentration reached a maximum of 0.97 mg/L after 3 days. Furthermore, a significant increase in REE concentration was observed after 10 days (Figures 6a and 9 *A. niger*). *A. niger* CECT2807 can grow using monazite as sole phosphate source, reaching 1.37 mg/L of REE (Figure 9 Minimal medium). Consequently, the minimal medium favors the fungal dissolution of REE and seems to avoid the notable precipitation of oxalates observed when *A. niger* was grown in a rich medium. Nevertheless, a slower growth was observed when the fungus grew in minimal medium and monazite particles remained unreacted after 14 days (Figure 8d). REE concentration in the non-inoculated flask reached a maximum of 0.78 mg/L (Figure 9, unsterilized monazite). Concentrations of REE in the leachate when *A. niger* was inoculated onto non-sterilized monazite was 0.83 mg/L (Figure 9, *A. niger* unsterilized monazite). Among all REE leached, concentrations of cerium were the greatest, followed by Nd and La. When the native consortia and *A. niger* were combined, a syntrophic effect between both populations leading to a higher amount of REEs would have been expected. Although the results were only slightly improved, the REE precipitation was significantly avoided.

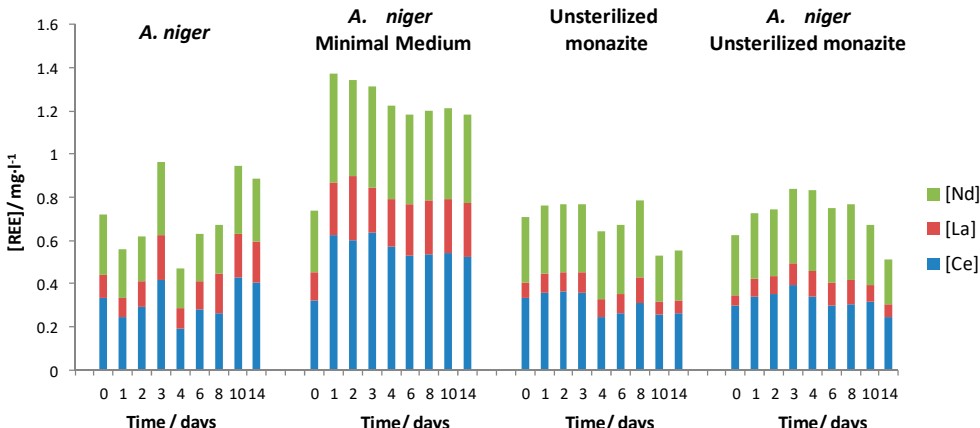

**Figure 9.** Total concentration of rare earth elements, cerium (Ce), lanthanum (La), and neodymium (Nd) bioleached during the experiments under different conditions.

Previous works have determined remarkable alterations in the natural microbial populations during bioleaching of monazite ores. The existence of native *Firmicutes* on the monazite seems to have significantly contributed to the increase in REE leaching observed when using non-sterilized monazite [38].

The leaching yields and rates in bioleaching are usually lower in comparison to chemical leaching using strong (in)organic acids or chelants. Sulfuric acid is often preferred to other inorganic acids because it dissolves fewer impurities, such as Ca and Sr [6]. However, bioleaching offers some advantages over chemical leaching in the case of certain low-grade REE ores; especially, the secondary sources. Bioleaching of waste phosphors and cracking catalysts using biogenic gluconic acid presented

higher efficiency than chemical leaching using pure organic acids at even higher concentrations, due to the presence of other metabolites in the leaching agent [39].

Techno-economic and life cycle analysis indicated that a bioleaching plant using agricultural or food wastes is an economical and environmental viable alternative for future REE recovery [40]. The use of waste products as alternative substrates for microbial metabolism may be more profitable and cleaner for several impact categories, with the same leaching efficiency but remarkably reduced processing costs and impacts.

## 4. Conclusions

This work has shown that valuable metals from mineral phosphates were mobilized using *A. niger* CECT2807. The phosphorous leaching rates were aluminum phosphate > turquoise > monazite. According to our results, an important amount of copper was recovered from turquoise. *A. niger* leached also REE from monazite, obtaining the best results when the fungus was cultivated on minimal medium with the mineral as the sole phosphorous source. The XRD results evidenced that *A. niger* was able to leach the copper aluminum phosphate hydroxide hydrate (turquoise) and the aluminum cerium phosphate hydroxide in monazite. Nevertheless, part of the leached copper and REE were precipitated as secondary minerals such as phosphates and oxalates. Bioleaching using phosphate solubilizing microorganisms could be an alternative to conventional methods for valuable metals extraction.

**Author Contributions:** Conceptualization, M.L.B.; Funding acquisition, J.A.M.; Investigation, L.C.; Methodology, J.A.M.; Project administration, F.G.; Resources, M.L.B.; Validation, F.G.; Writing—review and editing, L.C. and J.A.M. All authors have read and agreed to the published version of the manuscript.

**Funding:** This work was supported by the Complutense University of Madrid (project PR87/19-22648).

**Conflicts of Interest:** The authors declare no conflict of interest.

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
