# Peer review of "Bioleaching of Phosphate Minerals Using Aspergillus niger: Recovery of Copper and Rare Earth Elements"

_metals, doi:10.3390/met10070978_

Round 1
Reviewer 1 Report
This manuscript reports the bioleaching of Cu and REEs from phosphate minerals, which goes well with the aim and scope of the journal. However, the manuscript was not systematically and logically organized. This needs to be significantly modified or rewritten for publication in the journal.
- Introduction should be significantly revised. The background of research are not clearly presented in the introduction. Unfortunately, it is not certainly understood what the objective of this research is
- As for “ Bioleaching using phosphate solubilizing microorganisms could be an alternative to conventional methods for valuable metals extraction” in conclusion
This is not supported by the experimental results. As mentioned in the manuscript, the minerals of copper and REEs were transformed to the oxalate during bioleaching and their oxalates and phosphates are well known to little dissolve in water and even acidic solution. The formation of copper and REEs oxalates and phosphates rather limit the long-term dissolution of those elements.
Author Response
This manuscript reports the bioleaching of Cu and REEs from phosphate minerals, which goes well with the aim and scope of the journal. However, the manuscript was not systematically and logically organized. This needs to be significantly modified or rewritten for publication in the journal.
- Introduction should be significantly revised. The background of research are not clearly presented in the introduction. Unfortunately, it is not certainly understood what the objective of this research is.
Author's response
We would like to thank to the reviewer the comments to improve the manuscript. The introduction has been revised and more references have been included. Furthermore, the authors tried to improve and clarify the objective of the research.
- As for “ Bioleaching using phosphate solubilizing microorganisms could be an alternative to conventional methods for valuable metals extraction” in conclusion
This is not supported by the experimental results. As mentioned in the manuscript, the minerals of copper and REEs were transformed to the oxalate during bioleaching and their oxalates and phosphates are well known to little dissolve in water and even acidic solution. The formation of copper and REEs oxalates and phosphates rather limit the long-term dissolution of those elements.
Author's response
I agree with the reviewers comment; however, it has been reported that In a metallurgical process, REE-oxalate can be readily converted into REE-oxides through high temperature treatment and therefore may also serve as a precursor for other useful REE materials.
Kang, X., Csetenyi, L., and Gadd, G.M. (2019) Biotransformation of lanthanum by Aspergillus niger. Appl Microbiol Biotechnol 103: 981–993.
Authors have now included this comment in the manuscript.
Furthermore, this is a first approach to the bioleaching of copper and REE from phosphates and the results could be improved in further studies, for example, in continuous systems.
Reviewer 2 Report
The approach and the hypothesis that, according to my view, is considered for the research work is indeed of greater importance towards developing alternate methods for benign extraction of REEs. However, the authors fail to provide sound evidences that A. niger in this study have increased bioleaching of elements of interest. Following comments are a few examples as:
Intro contains many statements without providing any references.
There is no clear/specific aim for this study as random elements for extraction were chosen from minerals with different mineralogical/chemical structure.
Discussion were mainly done by hiding behind references. For example authors mentioned that organic acids production is major mechanism for heterotrophic bioleaching. However, no data on organic acids production from this study were reported. Authors also mentioned acidifaction can be responsible however, fig 2 doesn't show any significance change and in some cases pH was increased!
It has been agreed that sterlization of mineral by autoclave change mineralogy of minerals, specially for elements with various oxidation states.
In results section of the manuscript, all of sudden authors talk about Ca, calcium oxalate and calcium phosphate while there is no relevant information were provided previously in the manuscript. Also this mineral wasn't studied for P solubuilization.
Line 231-232 authors claimed that REEs concetration began to increase after day 4 due to a shift on P whereas fig 1 doesn't show any significant change of P conc during this period.
Author Response
The approach and the hypothesis that, according to my view, is considered for the research work is indeed of greater importance towards developing alternate methods for benign extraction of REEs. However, the authors fail to provide sound evidences that A. niger in this study have increased bioleaching of elements of interest. Following comments are a few examples as:
Intro contains many statements without providing any references.
Author's response
Thank you very much for your comments and your help to improve the manuscript. Additional references has been provided following your suggestion.
There is no clear/specific aim for this study as random elements for extraction were chosen from minerals with different mineralogical/chemical structure.
Author's response
Several hundred phosphate minerals are known and the most important mineral is apatite. Apatites and poorly crystalline or amorphous phosphorite sediments are mined and treated with acids to obtain fertilizers. Phosphates have been scarcely mined to extract valuable metals. Nevertheless, many of the f-block elements are extracted from phosphate minerals, notably monazite. Turquoise mineral was considered in this study as a source of copper. Some copper companies do not consider turquoise valuable enough for sale and they crush and process the mineral for its metal content. The aim for the study has been improved in the text.
Discussion were mainly done by hiding behind references. For example authors mentioned that organic acids production is major mechanism for heterotrophic bioleaching. However, no data on organic acids production from this study were reported. Authors also mentioned acidification can be responsible however, fig 2 doesn't show any significance change and in some cases pH was increased!
Author's response
We did not measure the concentration of organic acids, but the production of gluconic, citric and oxalic acids by A. niger has been reported in previous studies. Furthermore, the biogenic formation of oxalic acid is evidenced by the precipitation of oxalates shown in X-ray diffractograms.
The pH of the medium was 5.3. This value changed when the minerals were added. The initial pH of the medium with turquoise was 4.9 and this value decreased up to 2.8 at day 6. The initial pH of the medium with monazite was 3.4 and this value decreased up to 2.2 at day 7. It is explained in the manuscript that " After 7 days, the pH values in the inoculated flasks with AlPO4 and turquoise experimented a sharp increase associated to the release of phosphorous in the aqueous phase due to proton consumption. In addition, turquoise is a hydrated basic phosphate, CuAl6(PO4)4(OH)3·5H2O, that raises the OH- concentration during its solubilization."
It has been agreed that sterlization of mineral by autoclave change mineralogy of minerals, specially for elements with various oxidation states.
Author's response
I agree with the reviewer's appreciation. We have previous experience with ferric compounds. Nevertheless, we have compared X-ray diffractograms before and after autoclaving and no mineralogical changes were observed.
In results section of the manuscript, all of sudden authors talk about Ca, calcium oxalate and calcium phosphate while there is no relevant information were provided previously in the manuscript. Also this mineral wasn't studied for P solubuilization.
Author's response
Authors have removed this paragraph following reviewers suggestion. Authors included theses lines because calcium oxalate is the most common form of oxalate associated with soils and leaf litter: Furthermore, calcium oxalate crystals are commonly associated with free-living, pathogenic, and plant-symbiotic fungi and are formed by the reprecipitation of solubilized calcium as the oxalate.
Line 231-232 authors claimed that REEs concetration began to increase after day 4 due to a shift on P whereas fig 1 doesn't show any significant change of P conc during this period.
Author's response
I think there is a misunderstanding in this sentence. Authors have modified the paragraph to clarify it. Authors stated that "After three days of bioleaching, the REE concentration reach a maximum (0.97 mg/l).... A significant REE loss was observed after four days that may be caused by the removal of REE from the aqueous phase due to processes such as re-precipitation…" Perhaps the shift in not so eye-catching because of the high amount of dissolved P during the AlPO4 bioleaching experiment; however, phosphorous concentration changed from 0.70 mg/l at day 9 to 8.80 mg/l at day 14. This change has been mentioned in the text. In addition, mineral concentration was 1% and monazite contents 1.42% of P while turquoise content is 7.3%.
Reviewer 3 Report
Review
Manuscript Number: metals-849107
Title: Bioleaching of phosphate minerals using Aspergillus 2 niger: recovery of copper and rare earth elements
In this paper, the authors have investigated the recovery of Cu and REE from phosphates using bioleaching process using fungus Aspergillus niger. They have examined different mineral phosphates, like aluminum phosphate (commercial), turquoise and monazite (natural minerals) and bioleaching under various conditions. Finally the authors have reached 8,81 mg/l of Cu and 1,37 mg/l of REE after bioleaching. It was also found that used fungus was involved in the formation of secondary minerals such as Cu and REE oxalates.
There are some questions and remarks to be answered:
- There is a lack of Graphical abstract .
- The authors should perform a proof reading of the text (some mistakes, typos, etc.).
- The authors should indicate in the introduction what they have exactly achieved.
- Authors should describe all used reagents because some materials are missing and their parameters given by producer.
- There is a lack of information, concerning the applied method of error analysis (only 3 measurements?), method accuracy and reproducibility, for instance, in Figure 1-3, 6 and 9.
- The symbols of used parameters, such as: redox potential should be presented in the study.
- There is no description of obtained results for monzanite in chapter 3.1.
- There is a lack of information regarding the experiment time in Figure 4 and 7.
- The description of the results of Figure 9 should be supplemented with references to it.
- Authors should add equations of specific reactions that may occur during bioleaching.
- Did the authors consider the effect of other elements during the bioleaching process by testing the different compositions of tested samples?
- How did the authors determine the optimal conditions for conducting the experiment? Did they consider changes in temperature, pressure or use of neutral gas?
- There is a lack of described apparatus for determining pH and potential, its accuracy and repeatability.
- Authors should compare the efficiency of proposed technique with conventional methods and comment on its profitability.
- What about the efficiency of proposed method in industrial scale?
- Authors should standardize literature according to the publisher's template and supplement literature with additional articles
I would suggest this paper for publication, after rewriting it and introducing the proposed amendments.
Author Response
There are some questions and remarks to be answered:
1. There is a lack of Graphical abstract .
Author's response
The Graphical abstract is now included.
2. The authors should perform a proof reading of the text (some mistakes, typos, etc.).
Author's response
The text has been carefully read by a native English speaker.
3. The authors should indicate in the introduction what they have exactly achieved.
Author's response
Thank you for your comment. We have improved the introduction indicating the achievements and clarifying the aim of the study.
4. Authors should describe all used reagents because some materials are missing and their parameters given by producer.
Author's response
More information about reagents and equipments has been included in the manuscript following your suggestion.
5. There is a lack of information, concerning the applied method of error analysis (only 3 measurements?), method accuracy and reproducibility, for instance, in Figure 1-3, 6 and 9.
Author's response
The required information has been included in materials and method section. Each experiment was performed in triplicate. Furthermore, each sample was analyzed by ICP-OES and the result for each element concentration was obtained as the mean of 3 replicates and RSD < 3%.
6. The symbols of used parameters, such as: redox potential should be presented in the study.
Author's response
The symbol EhAg/AgCl has been already included in the manuscript.
7. There is no description of obtained results for monazite in chapter 3.1.
Author's response
Thank you for your comment. We have now included the description of the obtained results for monazite in chapter 3.1.
8. There is a lack of information regarding the experiment time in Figure 4 and 7.
Author's response
Samples for XRD were collected at the end of the experiments after 14 days of reaction. Now the information has been included in the figure caption following the reviewerʹs suggestion.
9. The description of the results of Figure 9 should be supplemented with references to it.
Author's response
Following the reviewer's suggestion, references to Figure 9 have been included in the description of the results.
10. Authors should add equations of specific reactions that may occur during bioleaching.
Author's response
The possible equations trough which bioleaching may occur have been included in the manuscript.
11. Did the authors consider the effect of other elements during the bioleaching process by testing the different compositions of tested samples?
Author's response
The aim of this work is the study of the dissolution of Cu and REE from phosphate minerals and the analysis has been focused on these elements. Furthermore, 5 elements were analyzed in monazite experiments, as well as pH and redox potential, representing a high sample consumption.
12. How did the authors determine the optimal conditions for conducting the experiment? Did they consider changes in temperature, pressure or use of neutral gas?
Author's response
It has been extensively reported that A. niger grows under aerobic conditions. Temperature was found to influence the growth of the isolated fungi. The optimum temperatures for growth of the Aspergillus species are 30 and 35°C. Only A. fumigatus appeared to have been influenced by incubation temperature of 40°C. Furthermore,the bioleaching of minerals at high pressure is not the aim of the study.
13. There is a lack of described apparatus for determining pH and potential, its accuracy and repeatability.
Author's response
The information about the pHmeter has been included in the manuscript. It is a pH meter Crison Basic 20 and sensitivity ˃98%.
14. Authors should compare the efficiency of proposed technique with conventional methods and comment on its profitability.
Author's response
A paragraph about some bioleaching processes that are profitable has been included in the manuscript.
15. What about the efficiency of proposed method in industrial scale?
Author's response
A comment about previous techno-economic and life cycle analyses comparing bioleaching and chemical leaching has been included.
16. Authors should standardize literature according to the publisher's template and supplement literature with additional articles
Author's response
New references have been included in the manuscript. MDPI template provided by Web of Science Group (Endnote) has been used for literature.
Round 2
Reviewer 2 Report
Authors argued that they have previous experience with ferric compounds. Nevertheless, they have compared X-ray diffractograms before and after autoclaving and no mineralogical changes were observed. I would suggest to include this in the manuscript so readers could follow the reasoning behind autoclaving.
Authors agreed they haven't measured organic acid concentration but it has been reported in the previous studies. More direct statement (e.g., based on previous studies) when such references were used is needed.
Please make sure consistency over the manuscript. For example some minor errors such as Table 37 (284) and when A. in capital bold (291).
Reviewer 3 Report
Accept in present form after revision.
Author Response
Thank you very much.